# Mechanistic insights into triclosan-induced hepatotoxicity: A network toxicology and molecular docking approach

Fuzhi Liu[1], Yanyan Zhao[2], Dandan Zhu[1], Jingwen Wang[1], Zhaocheng Zhuang[1], Yangjia Chen[1], Yuan Su[1], Zhuote Tu[1]*

1 Department of Preventive Medicine, School of Public Health, Quanzhou Medical College, Quanzhou, China, 2 Department of Nursing, Quanzhou First Hospital, Quanzhou, China

☉ These authors contributed equally to this work.
* 75582394@qq.com

## Abstract

### Background

Triclosan (TCS) is an artificially synthesized broad-spectrum antimicrobial agent, which is widely used in personal care products. It is a new endocrine disruptor and has potential health hazards to human body.

### Objective

Based on network toxicology and molecular docking technology, the compounds that may cause hepatotoxicity in triclosan were predicted and the mechanism was discussed.

### Method

From April to May 2025, the targets of triclosan were identified using databases such as STITCH, CTD, Swiss Target Prediction, and TargetNet. Additionally, gene targets associated with liver toxicity were identified from the GeneCards and OMIM databases. The intersection of triclosan-targets and liver toxicity-related gene targets was used to identify candidate targets. Using the String platform, a protein interaction network was constructed for these candidate targets to identify core functional modules within the network. The candidate targets were analyzed for GO and KEGG enrichment using DAVID, and a triclosan-liver toxicity-target pathway network was constructed using Cytoscape 3.10.1 software. Network topology analysis was conducted to screen for key components and targets. Finally, molecular docking was performed on the core targets using CB-Dock2.

**Data availability statement:** All relevant data are within the paper and its Supporting Information files.

**Funding:** This study was financially supported by the Education and Scientific Research Project for Young and Middle-aged Teachers of Fujian Province in the form of a grant awarded to FL (JAT251339). No additional external funding was received for this study. The funders had no role in study design, data collection and analysis, decision to publish, or preparation of the manuscript.

**Competing interests:** The authors have declared that no competing interests exist.

## Results

683 candidate targets for liver toxicity caused by triclosan were identified. The core targets for liver toxicity from triclosan production include *TP53, EGFR, AKT1, IL6, JUN,* and *FN1*. Molecular docking analysis shows that the binding free energy of triclosan with these core targets is less than −5.5 kcal/mol. The comprehensive analysis results showed that the liver damage caused by triclosan was mainly related to the activation of Pathways in cancer, Endocrine resistance, AGE-RAGE signaling pathway in diabetic complications, hepatitis B, and lipid and atherosclerosis signaling pathways.

## Conclusion

The potential targets and molecular mechanisms of triclosan (TCS) induced liver injury were investigated, and 6 key targets and 5 pathways were identified, providing a new paradigm for evaluating the health risks caused by environmental pollutants.

## Introduction

Triclosan (TCS) is a synthetic broad-spectrum antimicrobial agent that is widely used in personal care products such as soaps, toothpastes, mouthwashes, and antiseptic hand soaps, as well as for disinfecting and antimicrobial treatment of textiles [1]. With the increasing use of TCS, its environmental exposure shows an increasing trend, and can be detected in external environment such as water, air and soil [2–4]. The widespread application of TCS-containing antibacterial hand sanitizers during the COVID-19 pandemic has further heightened human exposure, underscoring the need to investigate its toxic mechanisms and potential therapeutic strategies [5].

As a lipophilic compound (log Kow = 4.8) with negligible water solubility but high solubility in organic solvents and alkaline solutions, TCS exhibits a strong tendency to accumulate in biological systems [6]. Its presence has been consistently documented in human matrices—including blood, urine, breast milk, amniotic fluid, adipose tissue, liver, and brain—following exposure through dietary and environmental pathways. This bioaccumulative behavior, particularly in lipid-rich tissues, underlies growing concerns regarding its potential health impacts [7–11]. Owing to its persistence and bioaccumulative potential, TCS presents a high exposure risk and has been associated with a range of adverse health effects, such as impaired fertility and development, behavioral abnormalities, embryotoxicity, hepatotoxicity, and intestinal toxicity [12–16]. While epidemiological studies have primarily focused on developmental toxicity, the molecular mechanisms underlying TCS-induced liver toxicity, particularly its role in hepatocellular carcinoma development, remain incompletely understood. According to human toxicological data, TCS has potential human carcinogenicity [17]. However, TCS has shown carcinogenicity only in mice and in some tissues (e.g., liver and colon), which according to the evaluation method of the European Chemicals Agency (ECHA), can only be classified as having limited evidence of carcinogenicity.

Therefore, TCS cannot be classified as a carcinogen at present. It should be noted that if future assessments ultimately exclude the possibility of TCS as a human carcinogen, caution must still be exercised with in view of the established carcinogenicity of its transformed products (dioxins, chloroform and aniline) [18].

The liver plays a central role in metabolism and detoxification. Due to its lipophilic nature, long-term exposure to TCS may have adverse health effects on the liver. More and more evidence suggests that TCS may disrupt liver metabolic function, aggravate liver fibrosis, accelerate the development of hepatocellular carcinoma through lipid metabolism disorder, oxidative stress, release of inflammatory factors, apoptosis, necrosis, fibrosis and metabolic dysfunction [19–22]. Studies have shown that TCS exposure significantly disrupts gut microbiota homeostasis, leading to excessive production of lipopolysaccharide (LPS). Furthermore, TCS increases intestinal permeability by reducing mucus secretion and decreasing tight junction protein expression, which further promotes LPS accumulation. The buildup of LPS in the bloodstream triggers inflammatory responses through the Toll-like receptor 4 (TLR4) pathway, ultimately causing liver damage [23]. Furthermore, TCS can induce liver injury by upregulating the TLR4-Myd88-TRAF6 signaling pathway, activating the NF-κB cascade, and initiating the NLRP3 inflammasome pathway [24]. Oxidative stress is recognized as a major contributor to diverse pathological processes. The overproduction of reactive oxygen species activates multiple transcription factors, thereby inducing immune and metabolic dysregulation. In the context of TCS-induced hepatotoxicity, peroxisome proliferation mediated through the peroxisome proliferator-activated receptor α (PPAR-α) signaling pathway has been identified as a key mechanism, ultimately leading to the inhibition of cellular proliferation [24,25]. Furthermore, TCS can influence m6A methylation through abnormal miR30b expression, leading to hepatic lipid metabolism disorders [26]. Triclosan induces zebrafish liver injury via the abnormal expression of miR-125 regulated by the PKCα/Nrf2/p53 signaling pathway [27].

Network toxicology provides a powerful framework for predicting compound toxicity by constructing compound-target-pathway-disease networks through bioinformatics platforms, enabling mechanistic insights before extensive in vivo validation [28]. Despite these advances, the molecular mechanisms underlying TCS-induced liver toxicity, particularly its role in hepatocellular carcinoma development, remain underexplored in existing research. Based on the above research results, this study explored how triclosan exposure affects liver injury and its molecular mechanisms through network toxicology and molecular docking technology, laying the foundation for subsequent research.

## Materials and methods

### Collection of Triclosan targets

The simplified molecular linear input specification (SMILES) number and SDF file of the chemical structure of triclosan were obtained from PubChem database for accurate identification of compound targets. The potential targets of triclosan were identified by searching the Swiss Target Prediction (http://swisstargetprediction.ch/), STITCH (http://stitch.embl.de/), Similarity Ensemble Approach (SEA, https://sea.bkslab.org/), TargetNet (http://targetnet.scbdd.com/) and Comparative Toxicogenomics Database (CTD, https://ctdbase.org/) databases, with selection criteria based on established network toxicology principles for comprehensive target identification, and limiting the species to "Homo sapiens". Predictions were filtered using stringent, database-specific criteria: with a probability cutoff of > 0.1 for SwissTargetPrediction, a significance level of P < 0.05 for SEA, a probability threshold of >0.6 for TargetNet, and a minimum inference score of ≥1 for CTD [29–32]. After merging the targets and removing the duplicates, the list of triclosan-related targets was finally integrated to form the chemical target library [33].

### Screening for hepatotoxicity-related targets

The Online Mendelian Inheritance in Man (OMIM, https://omim.org/) and GeneCards (https://www.genecards.org/) databases were comprehensively searched using terms such as liver toxicity, liver injury, liver dysfunction, liver damage, and

hepatotoxicity [34]. Specifically, the target genes were identified by using the precise matching method in OMIM database; the target genes with "relevance score" higher than 10 identified in GeneCards database were screened out to identify the disease targets highly related to human liver toxicity. The targets from these databases were merged and duplicates were excluded to identify potential relevant targets for hepatotoxicity. Finally, the intersection with the triclosan chemical target library was determined to identify the potential targets of triclosan-induced hepatotoxicity.

### Enrichment analysis of gene functions and pathways of target proteins

Gene ontology (GO) and Kyoto Encyclopedia of Genes and Genomes (KEGG) enrichment analysis were performed using DAVID (https://david.ncifcrf.gov/) database to explore the molecular pathways involved in the potential target genes of triclosan-induced liver toxicity [35]. The analysis covers the three GO categories (biological processes, cellular components and molecular functions) and KEGG pathways, aiming to reveal the biological role and molecular mechanism of the identified genes. Significantly enriched functional pathways were identified according to the standard of false discovery rate (FDR) < 0.05. In addition, KEGG enrichment analysis was performed on the core targets.

### Protein interaction analysis and core target screening

In order to further explore the core targets of triclosan-induced liver toxicity, we imported the cross-target genes of triclosan and liver toxicity into STRING 12.0 database to construct a protein-protein interaction (PPI) network. When the selected organism is "Homo sapiens" and the interaction score is "highest confidence (0.900)", the edges, nodes and degree values of these target genes are detected to construct the PPI network [36]. The darker the node color, the greater its degree value, which means that the target is more closely interacting with other targets. Then, the three plug-ins of CytoNCA, CytoHubba and MCODE in Cytoscape software (version: 3.10.1) were used to screen out the core targets. The selection criteria are as follows: CytoNCA sets Betweenness (BC), Closeness (CC) and Eigenvector (EC) greater than their respective medians, Degree (DC) greater than 4 times their median; CytoHubba selects the top 10 targets selected by MCC algorithm; MCODE selects the target set with the highest subnetwork score [37]. The intersection of the targets selected by these three methods was combined with repeated items to determine the core target of triclosan toxicity to liver.

### Molecular docking

The molecular structure of triclosan was obtained from PubChem database. In the Protein Data Bank (PDB, https://www.rcsb.org/), the three-dimensional structures of core target proteins with a degree value greater than 30 were downloaded from the website, and the screening criteria were: experimental source, experimental method was X-RAY Diffraction, Homo sapiens, lowest resolution and the latest year. Molecular docking was carried out using the online server Cavity-detection guided Blind Docking (CB-Dock2, https://cadd.labshare.cn/cb-dock2/php/index.php) [38]. For key proteins and ligands, missing atoms and hydrogen atoms were repaired, water molecules and other heteroatoms were removed, and the highest scoring binding sites were selected. Important protein structures with optimal conformation are visualized using an interactive NGL viewer.

### Gene expression validation

The microarray data set (GSE169072) was obtained from the Gene Expression Omnibus database to analyze the changes in gene expression profile of HepaRG cells, a primary liver cell replacement model, after 72 hours of treatment with 0.050 mM triclosan. Specifically, the results of the data set were presented in two groups: the control group (CTL: GSM5176060, GSM5176061, GSM5176062, n = 3) and the exposure group (0.050 mM triclosan, TRI: GSM5176072, GSM5176073, GSM5176074, n = 3). The GEO2R tool based on the Limma package was then used to analyze the GSE169072 dataset to identify differential expressed genes (DEGs) [39]. In the GEO2R tool, the P value and FDR of

DEGs were calculated using the t-test and the Benjamin-Hochberg method. The DEGs screening was set according to the following two screening criteria: $P < 0.05$ and $|log2FC (Fold Change)| \geq 0.26$.

### Key pathway screening

Functional pathways related to TCS-induced liver toxicity obtained from network toxicology and differentially expressed genes induced by TCS stimulation of HepaRG cells (all selected in the top 20). We screened for intersection pathways involved in the mechanism of liver injury caused by TCS in network toxicology and gene expression validation using UpSet plot analysis. Ultimately, these key pathways were identified as playing an important role in the hepatotoxicity caused by TCS.

## Results

### Targets of TCS-ignited hepatotoxicity identified

This study initially screened 3,533 TCS target genes using the Swiss Target Prediction, STITCH, SEA, TargetNet, and CTD databases. It then identified 2,501 highly liver toxicity-related targets using the OMIM and GeneCards databases. After integrating and deduplicating these target sets, 683 overlapping targets were identified, which may be potential targets for TCS-induced liver damage and represent a substantial intersection that suggests biologically relevant mechanisms (Fig 1 and S1 File).

### Functional enrichment analysis of TCS-induced liver injury targets

We conducted an enrichment analysis of 683 potential target genes to investigate the mechanism by which TCS induces hepatotoxicity. GO analysis identified 1216 BP, 149 CC, and 307 MF, with FDR < 0.05 as the significance threshold. The top 10 Gene Ontology (GO) entries with the lowest false discovery rate (FDR) in each category were selected for visualization (Fig 2 and S1 Table). In the biological process (BP) category, the target genes are primarily enriched in 'response to xenobiotic stimuli,' 'positive regulation of gene expression,' and 'positive regulation of transcription by RNA polymerase II.' In the cellular component (CC) category, 'extracellular exosome,' 'extracellular space,' and 'extracellular region' are significantly enriched, suggesting that these targets may play a role in cell signaling. The molecular function (MF) category

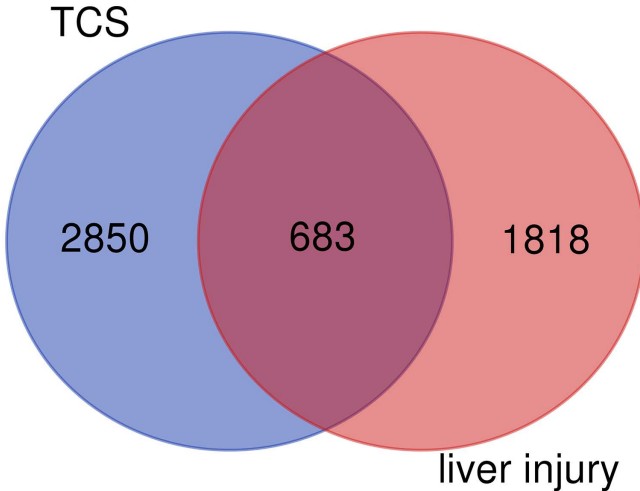

**Fig 1. The Venn diagram illustrating the targets of TCS and liver injury.**

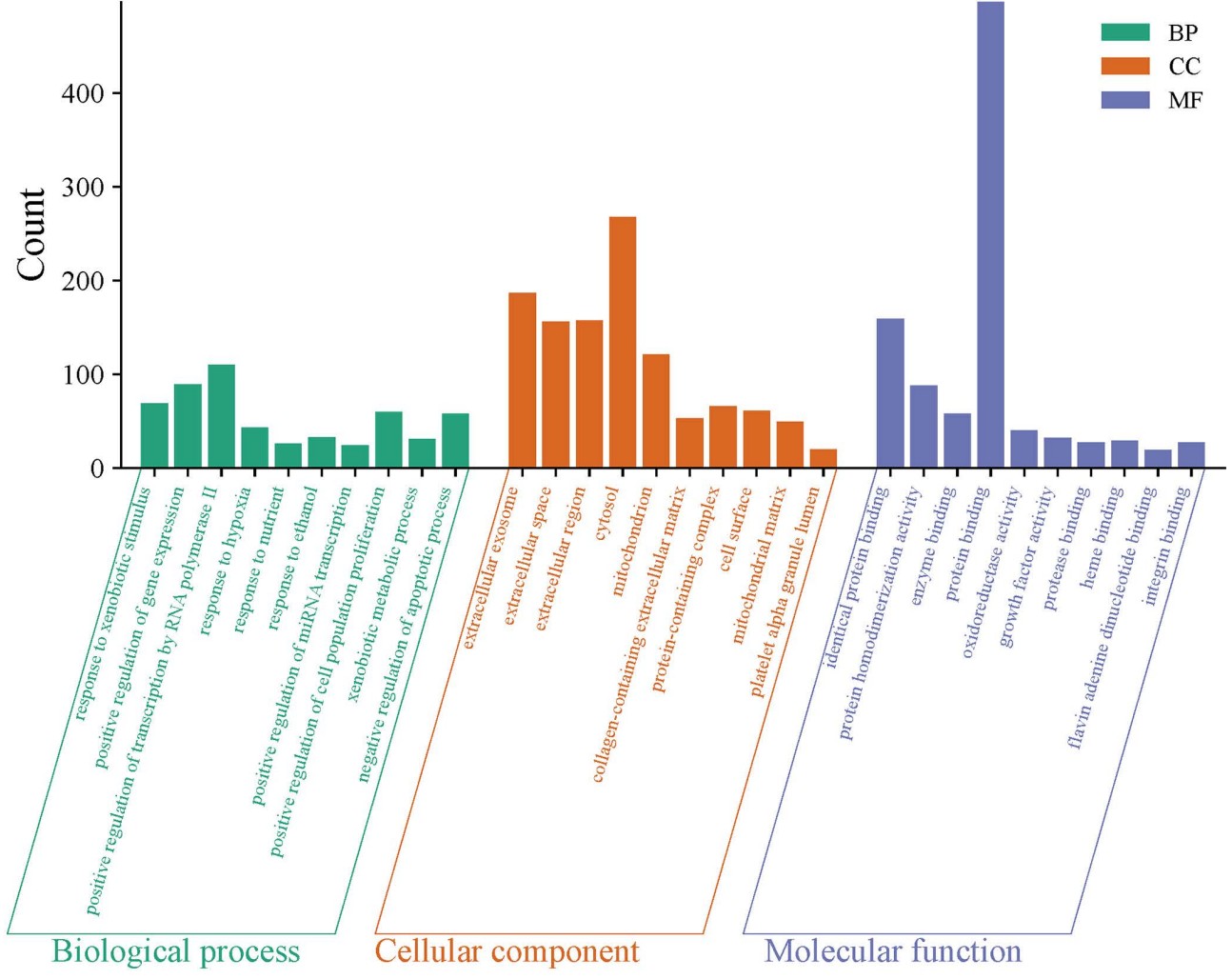

**Fig 2. Top 10 GO terms of the target genes in the GO enrichment analysis.**

shows significant enrichment in 'identical protein binding,' 'protein homodimerization activity,' and 'enzyme binding,' indicating that these targets may be involved in intercellular signal transmission. A total of 205 KEGG pathways were enriched, with FDR<0.05 as the significance threshold, and the top 20 were selected for visualization. These targets were mainly related to "Pathways in cancer", "AGE-RAGE signaling pathway in diabetic complications", "Lipid and atherosclerosis", "PI3K-Akt signaling pathway", and "Non-alcoholic fatty liver disease" (Fig 3 and S2 Table).

## Identification of core targets using Cytoscape

In order to further identify the potential key targets of liver damage caused by TCS exposure, we constructed a Protein-Protein Interaction (PPI) network of the above cross-potential targets by setting the highest confidence level (0.900). After removing some disconnected points, the network contains 566 nodes and 1844 edges, and the degree range of nodes is 1−55, with an average node degree value of 6.52 (Fig 4). It is worth noting that we used three Cytoscape plug-ins to screen for core targets. CytoNCA method screened out 20 potential core targets, MCODE method screened out 13 potential core targets, and CytoHubba method screened out 10 potential core targets. Finally, 29 core targets related to TCS-induced

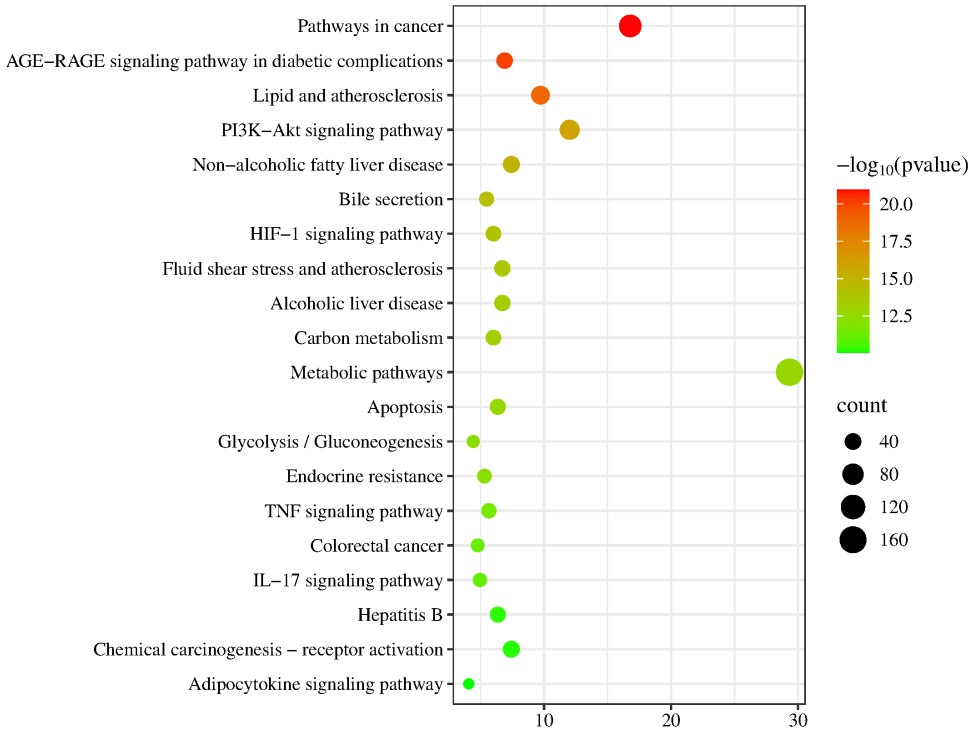

**Fig 3. Top 20 pathways of the target genes in the KEGG enrichment analysis.**

hepatotoxicity were identified by taking the union of the three, demonstrating high consistency across methods and supporting the biological relevance of the selected targets (Fig 5, S3 Table). Among these, interleukin 6 (IL6), tumor necrosis factor (TNF), and interleukin 1 beta (IL1B) appeared in all three screening methods. The top eight core targets, ranked by degree value, are tumor protein p53 (*TP53*), epidermal growth factor receptor (*EGFR*), AKT serine/threonine kinase 1 (*AKT1*), interleukin 6 (*IL6*), Jun proto-oncogene, AP-1 transcription factor subunit (*JUN*), fibronectin 1 (*FN1*), estrogen receptor 1 (*ESR1*), and tumor necrosis factor (*TNF*). All these core targets have a degree value of 30 or higher.

## Analysis of pathways of core targets

We conducted a KEGG enrichment analysis on 29 core targets using the DAVID database. A total of 139 pathways were enriched, with FDR < 0.05 as the significance threshold, and the top 20 pathways with the lowest FDR values were selected for visualization, resulting in a Sankey diagram of core target enrichment (Fig 6 and S4 Table). These core targets are primarily associated with Lipid and atherosclerosis, IL-17 signaling pathway, AGE-RAGE signaling pathway in diabetic complications, Pathways in cancer, and Amoebiasis. When comparing the top 20 pathways with potential targets, 10 pathways were found to be consistent: Pathways in cancer, AGE-RAGE signaling pathway in diabetic complications, Lipid and atherosclerosis, Non-alcoholic fatty liver disease, Endocrine resistance, TNF signaling pathway, IL-17 signaling pathway, Hepatitis B, and Chemical carcinogenesis-receptor activation. Notably, the rankings of the IL-17 signaling pathway and the TNF signaling pathway moved from 17th to 2nd and from 15th to 8th, respectively (S5 Table).

## Molecular docking study of TCS and core target proteins of liver injury

To investigate the interactions between TCS and eight core target proteins (*TP53, EGFR, AKT1, IL6, JUN, FN1, ESR1,* and *TNF*), we conducted molecular docking analysis. Using the online server CB-Dock2 to rapidly generate docking

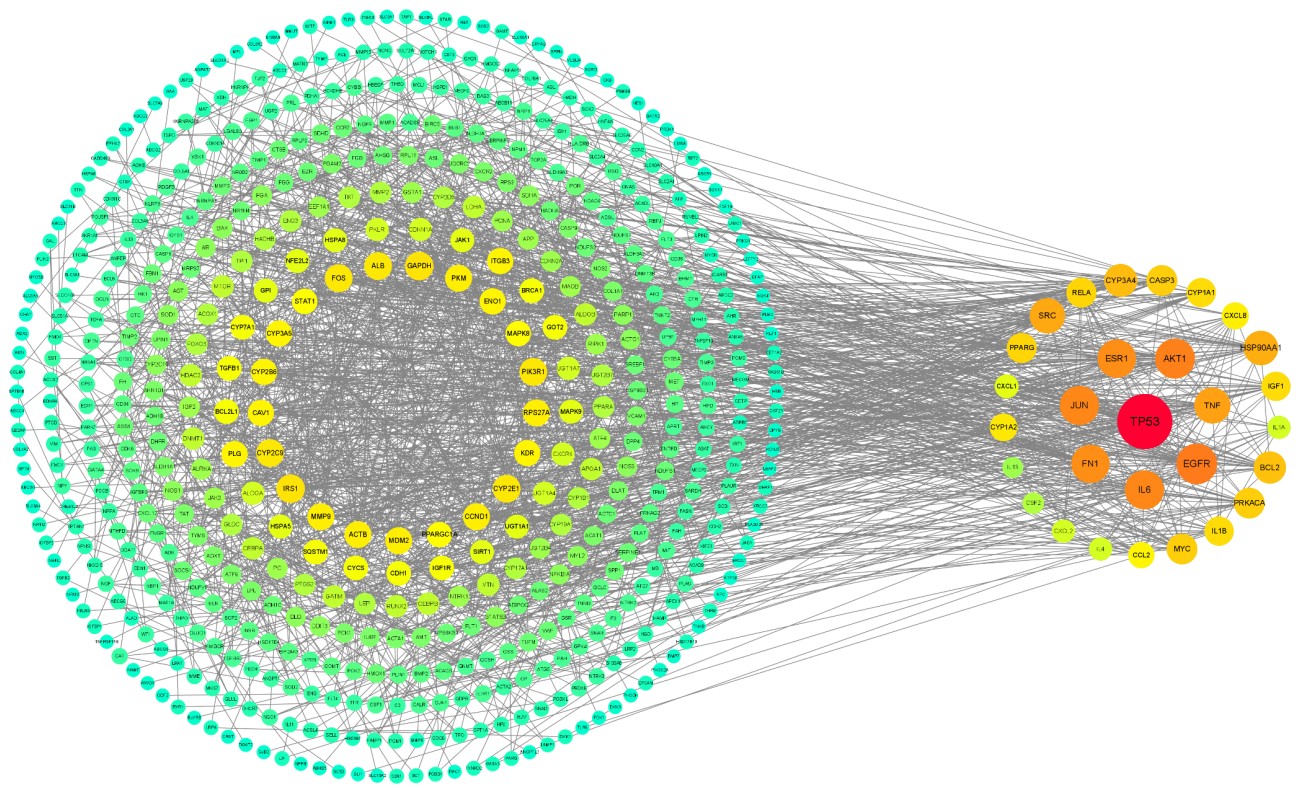

**Fig 4. The protein-protein network interactions of potential targets.**

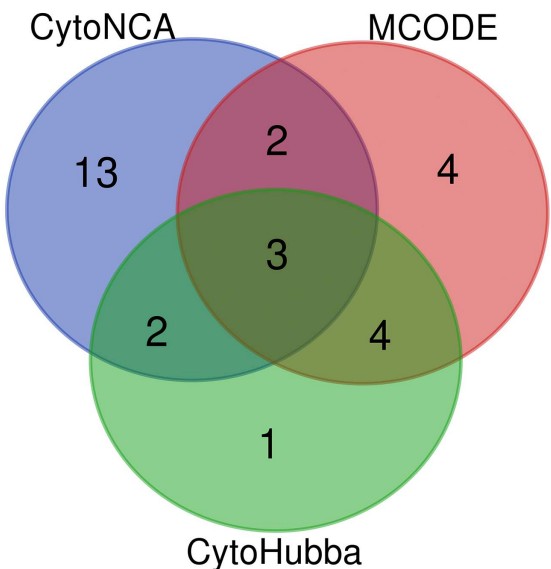

**Fig 5. Core genes identified using three Cytoscape plug-ins.**

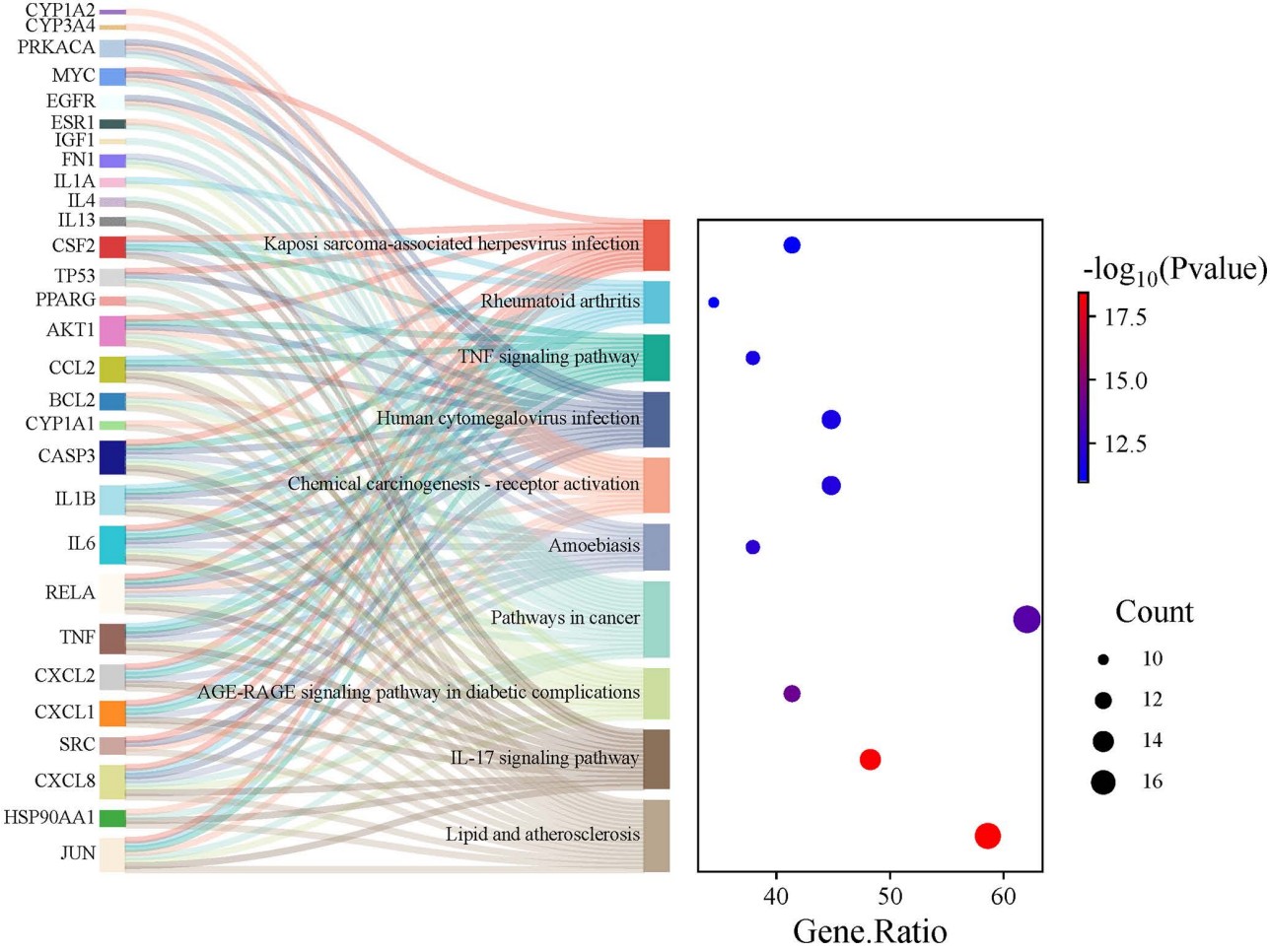

**Fig 6. Top 10 KEGG pathways for core targets Sankey and bubble diagrams.**

results, we found that the binding energies were −7.1, −7.7, −6.6, −5.5, −6, −6.5, −6.2, and −5.5 kcal/mol, respectively, indicating potential strong affinity between TCS and these core target proteins (Figs 7–14, S6 Table). Notably, the binding energy of TCS with each of these eight core target proteins was less than −5.5 kcal/mol, suggesting that TCS can potentially bind to each target protein with favorable binding affinity. Though these computational predictions require experimental validation, suggesting that these proteins play a crucial role in the molecular mechanisms of TCS-induced hepatotoxicity and liver damage. Among them, the lowest binding energy was observed between TCS and EGFR (−7.7 kcal/mol), indicating that TCS has the strongest binding affinity for EGFR among all targets.

## Results of gene expression validation

To verify the expression level of core genes, we drew a heat map of core genes (Fig 15, S7 Table and S2 File). Compared with the control group, the expression of *TP53* (log2FC=0.45, p = 0.001), *EGFR* (log2FC=0.44, p = 0.001), *AKT1* (log2FC=0.13, p = 0.002), *IL6* (log2FC=0.80, p = 0.008) and *JUN* (log2FC=0.28, p = 0.003) was significantly increased, while the expression of *FN1* (log2FC=−0.33, p = 0.028) was significantly decreased. *ESR1* (log2FC=−0.18, p = 0.213) and *TNF* (log2FC=−0.11, p = 0.155) were not significantly differentially expressed in this dataset under the specified experimental conditions (Fig 16 and S8 Table).

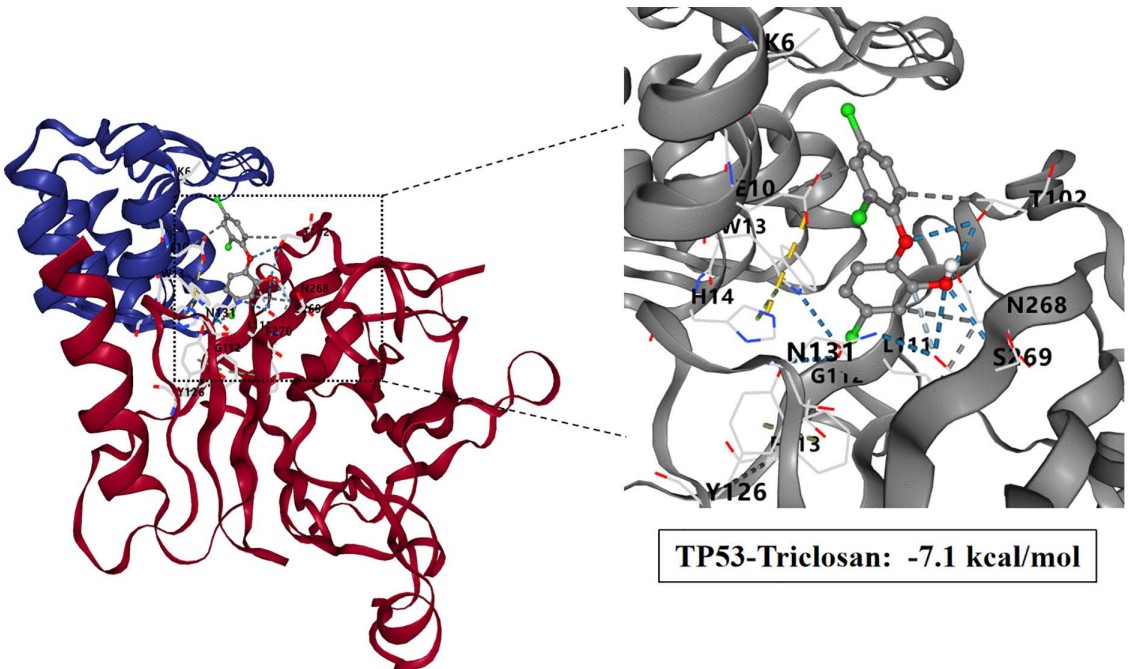

**Fig 7. Visualization of molecular docking results for triclosan with TP53.**

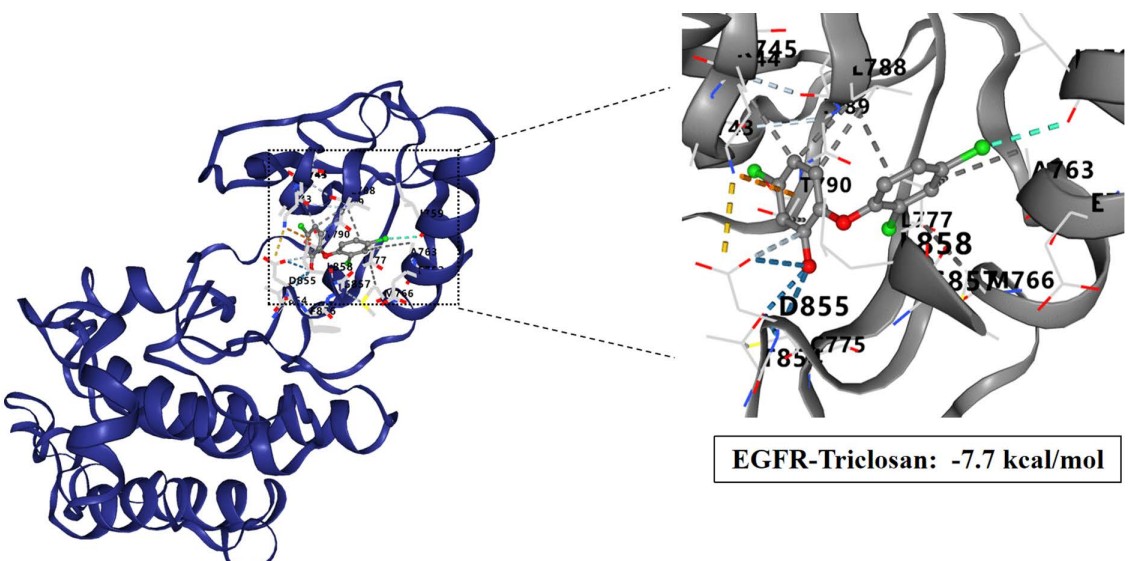

**Fig 8. Visualization of molecular docking results for triclosan with EGFR.**

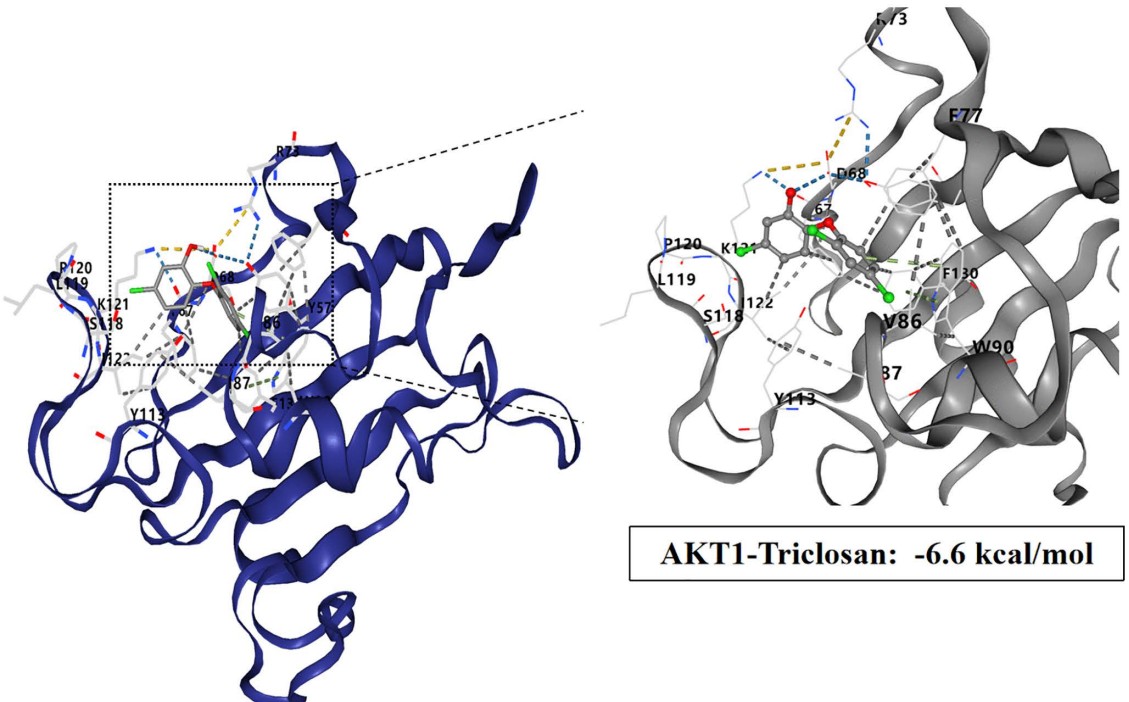

**Fig 9. Visualization of molecular docking results for triclosan with AKT1.**

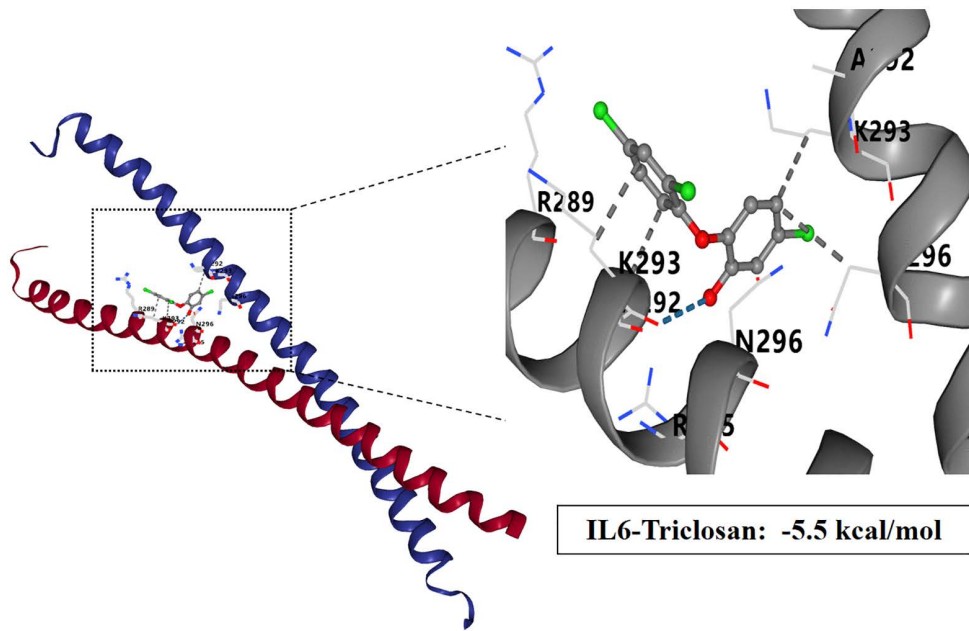

**Fig 10. Visualization of molecular docking results for triclosan with IL-6.**

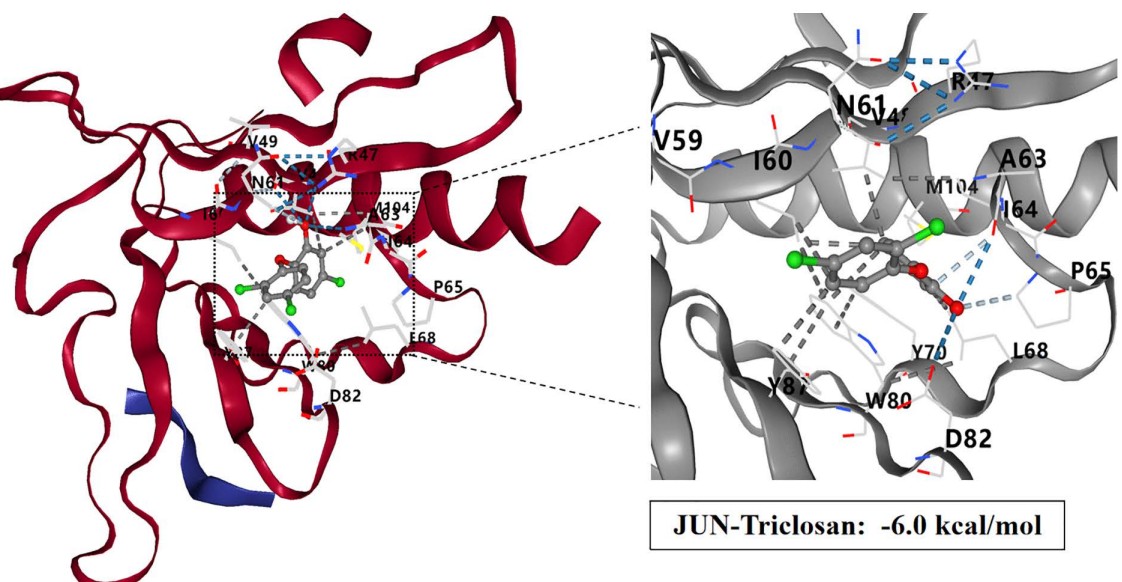

**Fig 11. Visualization of molecular docking results for triclosan with JUN.**

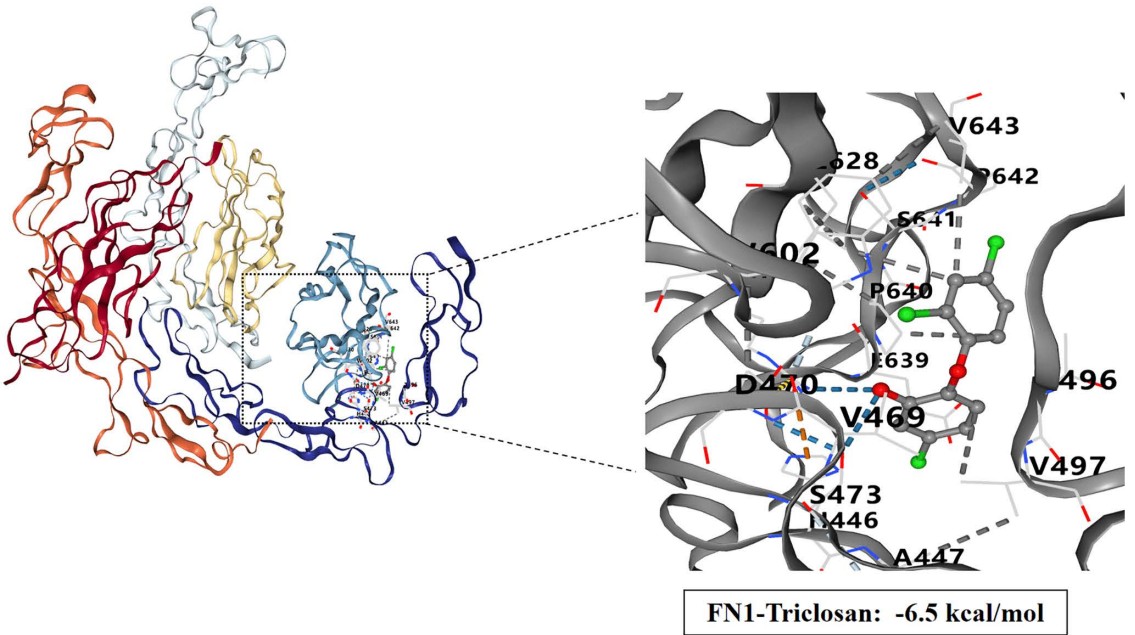

**Fig 12. Visualization of molecular docking results for triclosan with FN1.**

## Results of key pathways

To evaluate the robustness of our identification strategy, we assessed the consistency among the three target sets: the 683 potential targets, the 29 core targets, and the 8 validated targets. A substantial overlap was observed, particularly for pivotal targets such as *TP53, EGFR, AKT1,* and *IL6*, which were consistently identified across the analytical workflow.

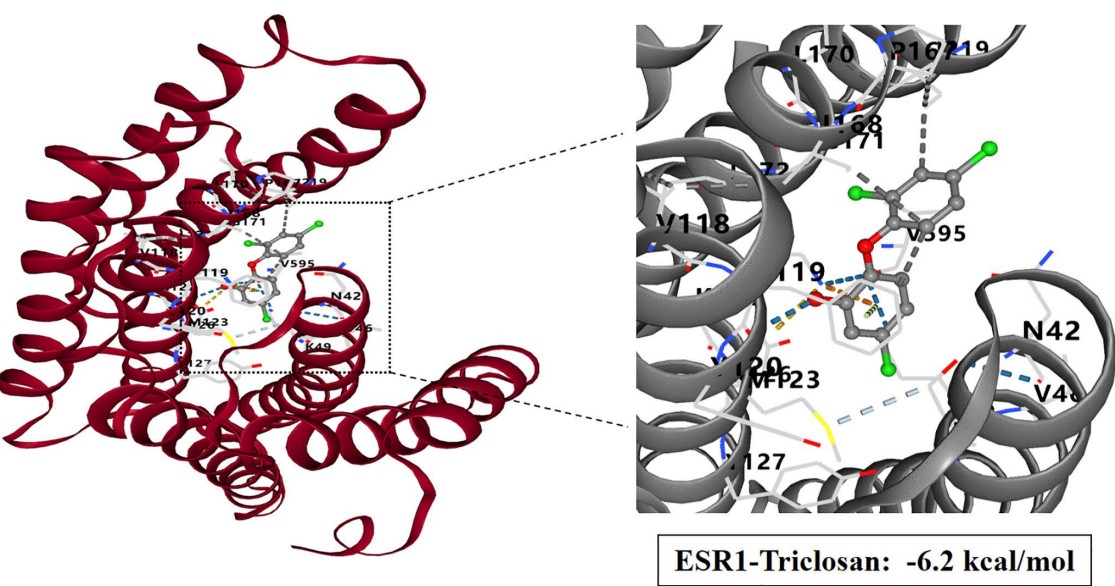

**Fig 13. Visualization of molecular docking results for triclosan with ESR1.**

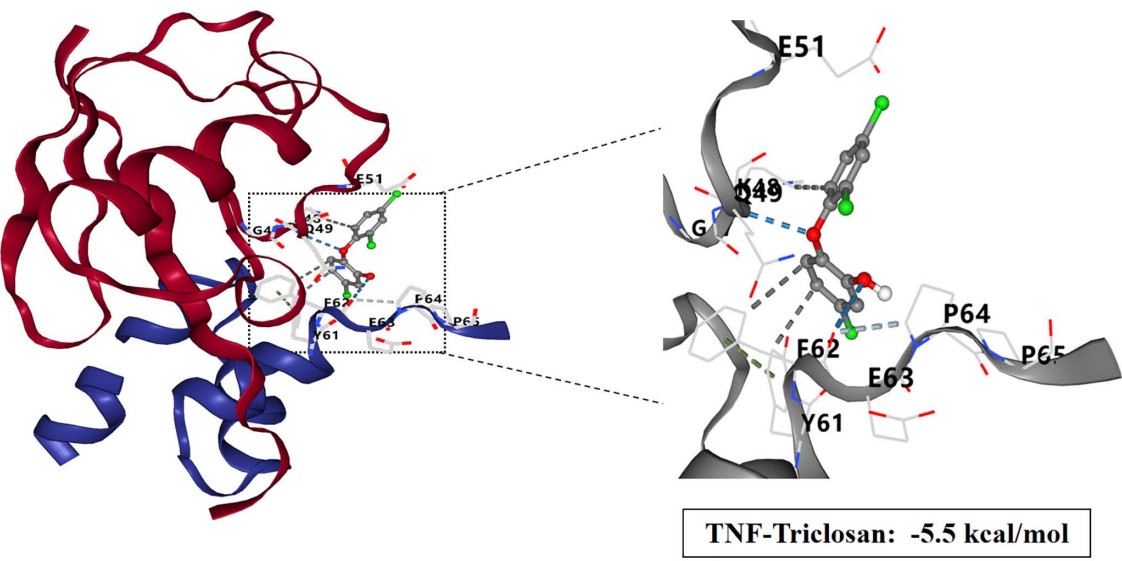

**Fig 14. Visualization of molecular docking results for triclosan with TNF.**

This convergence underscores the reliability of our multi-tiered screening approach. The UpSet plot analysis revealed five shared functional pathways: Pathways in cancer, Endocrine resistance, AGE-RAGE signaling pathway in diabetic complications, Hepatitis B, and Lipid and atherosclerosis, which were consistently identified across all target sets (Fig 17 and S9 Table). The high consistency observed is biologically meaningful, as it indicates that the network pharmacology screening effectively prioritizes targets and pathways central to the mechanism of TCS-induced hepatotoxicity, thereby validating our methodological framework.

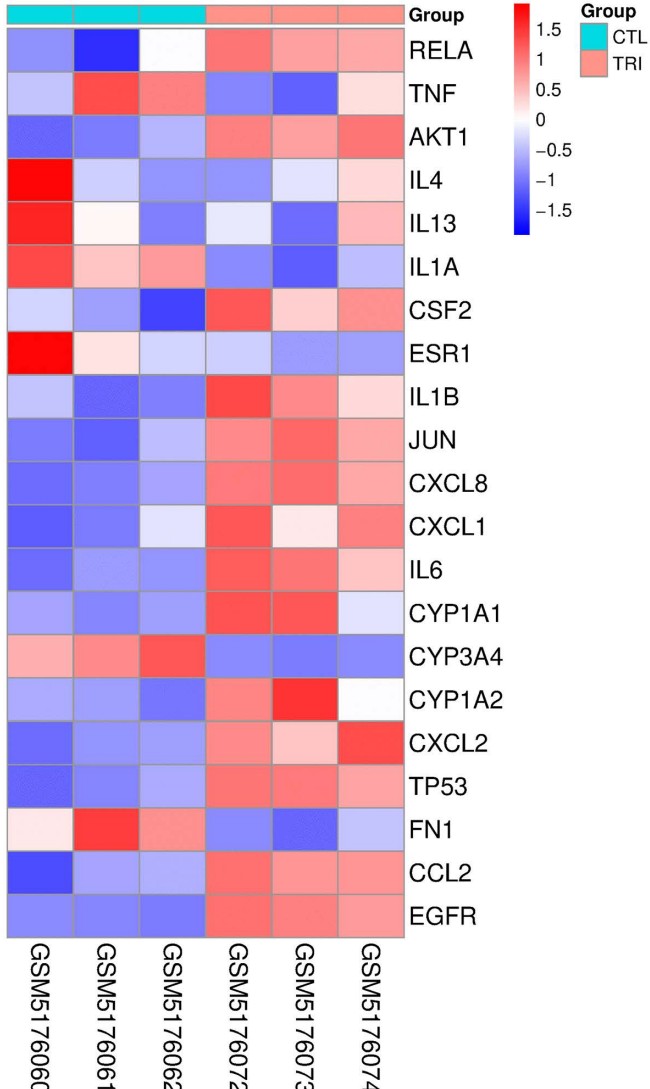

**Fig 15. Heat map of representative genes between control and TCS.**

## Discussion

This study employed network toxicology methodologies to integrate gene-target and compound data through cross-database analysis, identifying 683 potential targets associated with triclosan-induced hepatotoxicity. By constructing protein interaction networks using STRING and Cytoscape software, core target identification was achieved through multiple algorithms. Subsequent molecular docking and gene expression validation identified six key genes: *TP53, EGFR, AKT1, IL6, JUN,* and *FN1*, along with five critical signaling pathways: Pathways in cancer, Endocrine resistance, AGE-RAGE signaling pathway in diabetic complications, Hepatitis B, and Lipid and atherosclerosis.

As we all know, tumor development depends on the activation of oncogenes and the loss of function of tumor suppressor genes [40–41]. The TP53 gene is one of the most important tumor suppressor genes and a major participant in cancer formation [42]. As a transcription factor, p53 protein becomes active by forming tetramers, which can rapidly respond to

**Fig 16. Expression of eight key targets between control and TCS.**

many environmental stimuli and participate in biological processes such as DNA repair, senescence, cell cycle control, autophagy and apoptosis, ferroptosis, etc. Then it directly or indirectly activates a variety of genes and starts the TP53 signal pathway [43–45]. TP53 is the most frequently mutated tumor suppressor gene, with mutations present in over 50% of human tumors [46]. Triclosan, as a novel environmental pollutant, acts as an exogenous stress signal. These stress signals activate signaling pathways that modify the p53 protein through post-translational modifications such as phosphorylation and acetylation. This process releases p53 from its binding to the MDM2 protein, ultimately activating P53 [47]. Subsequently, p53 enters the nucleus and induces extensive expression of target genes [48]. Through this series of

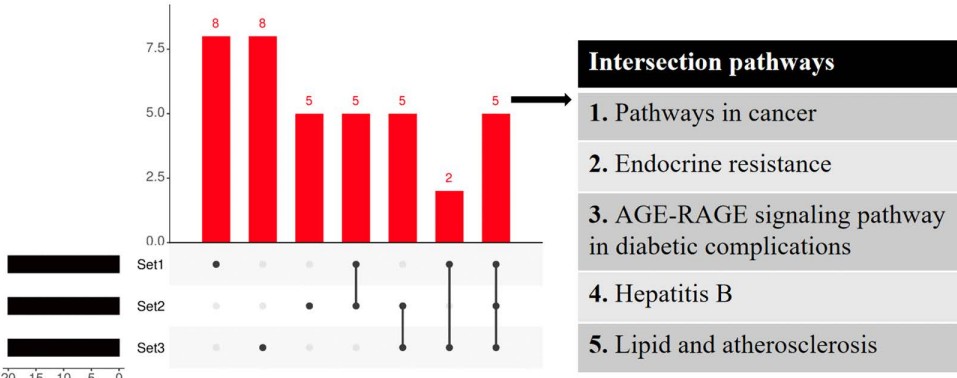

**Fig 17. Integrated analysis of the toxicity mechanisms of TCS.**

biological changes, liver toxicity is consequently induced. In this molecular docking study, the binding free energy of TP53 and triclosan was shown to be −7.1 kcal/mol. Studies have shown that fractions below −5.0 kcal/mol indicate potential binding possibilities, while fractions below −7.0 kcal/mol indicate strong binding capacity [49]. Thus, the binding of triclosan to TP53 plays an important role in inducing hepatotoxicity.

Furthermore, this study identified the Pathways in cancer pathway as a critical mechanism by which triclosan induces liver damage. As previously mentioned, p53 protein, a transcription factor, is known as the "guardian of the genome" due to its essential role in maintaining genomic integrity [50]. Numerous studies have demonstrated that p53 protein effectively participates in regulating cancer development through complex interactions with multiple signaling pathways. Research has identified the P53-P21-RB1-USP1-BRD4 axis as a pivotal mechanism in hepatocellular carcinoma progression. The study revealed a significant negative correlation between p53 protein levels and BRD4 protein levels in liver cancer tissues, with p53 demonstrating inhibitory effects on BRD4 stability. As the ubiquitination degrader of BRD4, USP1 is suppressed by p53 through the P21-RB1 axis. This USP1-BRD4 axis promotes hepatocyte proliferation by upregulating cancer-associated genes [51]. In addition, p53 can also inhibit mTor kinase and p53/p21 Cdk-interacting protein 1 (Cip1), activate NF-κB, stimulate Raf/MEK/ERK signaling pathway, inhibit p53/p21/p27 and p53/Bcl-2/Bax pathways in a variety of ways to affect cell proliferation, G2/M phase regulation and apoptosis and other biological processes [52–53].

Epidermal Growth Factor Receptor (EGFR), a glycoprotein belonging to the ErbB receptor family as a tyrosine kinase receptor, is located on the cell membrane surface. It is activated by ligand binding, including EGF and TGFα (transforming growth factor α) [54–57]. Upon ligand binding and activation, EGFR triggers homodimerization or heterodimerization and undergoes autophosphorylation at key cytoplasmic residues. The phosphorylated receptor recruits adapter proteins like GRB2, thereby activating complex downstream signaling pathways such as RAS-RAF-MEK-ERK, PI3 kinase-AKT, PLCγ-PKC, STATs, and NF-κB signaling [58–63]. Studies have confirmed that high or abnormal expression of EGFR is associated with tumor cell proliferation, angiogenesis, tumor invasion, metastasis, and inhibition of apoptosis [64–65]. Exposed to triclosan, the organism converts extracellular clues into appropriate cellular responses, activating EGFR and triggering a series of signaling cascades. Studies have shown that EGFR is the core gene of triclosan exposure, which is consistent with the results of this study [66].

Chronic inflammation serves as a common foundation for the development of numerous noncommunicable diseases, particularly diabetes, atherosclerosis, and tumors. Within the AGE-RAGE signaling pathway, Advanced Glycation End products (AGEs) bind to RAGE to activate signaling pathways such as NF-κB, thereby promoting the release of inflammatory mediators. This exacerbates cellular damage and inflammation, further accelerating tumor progression [67]. The PI3K-AKT signaling pathway is the most frequently activated pathway in human cancers. The transcription factors within this pathway

are highly regulated through multiple cross-interactions with several other signaling pathways, leading to dysregulated signal transduction. This promotes cell survival, growth, and progression through the cell cycle, ultimately driving cancer development [68]. In the IL-17 signaling pathway, upon binding to its receptor, IL-17 activates NF-κB, MAPK, and STAT pathways through the Act1-TRAF6-TAK1 complex. This induces the expression of genes encoding proinflammatory factors, chemokines, and matrix metalloproteinases, thereby driving inflammatory responses [69]. The strong consistency observed across the potential, core, and validated target sets enhances the credibility of our findings. The recurrent identification of key targets such as TP53 and IL6—along with pathways related to inflammation and carcinogenesis—through independent analytical approaches indicates that these are not methodological artifacts, but rather fundamental elements in TCS-induced hepatotoxicity. Notably, core targets including TP53, EGFR, AKT1, and IL6 are co-localized within IL-17 and TNF signaling pathways, suggesting crosstalk between inflammatory and oncogenic processes as a central mechanism. The elevated prominence of IL-17 and TNF signaling among core targets further supports their role in amplifying inflammatory responses that may synergize with carcinogenic pathways. Together, these network-level interactions provide a plausible mechanistic framework linking TCS exposure to the development of liver pathology.

This study has achieved two significant outcomes: First, it elucidated the potential molecular mechanisms of TCS hepatotoxicity, enhancing our understanding of the health impacts of environmental pollutants like TCS. Second, we have developed a rapid and comprehensive evaluation method that integrates bioinformatics, genomics, network toxicology, and molecular docking strategies. This approach not only assesses the toxicity of toxic chemicals and identifies their potential molecular mechanisms but also effectively addresses traditional toxicological evaluation limitations such as time-consuming processes, ethical concerns in animal testing, and limited predictive capabilities. These innovative methods significantly improve the efficiency, scope, and depth of toxicological screening, making it possible to evaluate numerous emerging environmental pollutants that have yet to be fully studied.

## Conclusion

This study employed network toxicology and molecular docking analysis to investigate potential targets and molecular mechanisms of triclosan-induced liver injury. Six hundred and eighty-three candidate targets were identified, with 29 core targets further validated through Cytoscape's learning algorithms. Genomic integration revealed six key targets and five biological pathways. The research demonstrated that TCS exerts hepatotoxic effects by regulating critical targets including TP53, EGFR, AKT1, IL6, JUN, and FN1. These interactions disrupt signaling pathways such as Pathways in cancer, Endocrine resistance, AGE-RAGE signaling pathway in diabetic complications, Hepatitis B, and Lipid and atherosclerosis, ultimately leading to carcinogenic effects, inflammatory responses, and apoptosis-induced liver damage, though these computational predictions require further experimental validation.

## Limitations

While our research has achieved certain results, several limitations should be noted. 1. Lack of experimental validation: The primary limitation of this study lies in the absence of animal models or in vitro experiments to directly validate the core target sites and signaling pathways predicted by network toxicology. Consequently, computational results remain at the hypothetical stage, failing to translate into experimentally substantiated biological conclusions. This undermines their reliability as direct support for preventive and therapeutic strategies. 2. Complexity of real-world exposure scenarios: The study failed to adequately simulate and explore exposure scenarios in actual environments. In practice, triclosan frequently coexists with other endocrine disruptors within complex mixtures, potentially generating synergistic or antagonistic effects that influence final hepatic toxicity. This limitation impacts the applicability and predictive value of findings in real-world contexts.3. Inherent bias and incompleteness of databases: Public databases relied upon (e.g., SwissTargetPrediction, CTD) may themselves suffer from data bias, coverage gaps, or outdated information. This directly compromises the comprehensiveness and accuracy of target predictions, constituting an inherent source of uncertainty in computational

toxicology approaches. Furthermore, inconsistent screening criteria and target prediction guidelines during database utilization can introduce predictive bias.4. Absence of individual variability factors (limitations of refinement): This study did not incorporate individual factors such as age and gender, which may significantly influence susceptibility to hepatic toxicity, into the analytical framework. This constrains understanding of risk differences across populations, rendering conclusions relatively generalized. Future research directions: The primary task is to functionally validate the key targets and signaling pathways identified by network toxicology through in vitro and in vivo liver toxicity experiments. Conducting epidemiological studies tracking the dynamic relationship between triclosan exposure and liver toxicity will provide a robust theoretical foundation for establishing population-based prevention strategies.

## Supporting information

**S1 Table. The top 10 Gene Ontology terms of 683 potential target genes.**
(DOC)

**S2 Table. The top 20 pathways of 683 potential target genes.**
(DOC)

**S3 Table. Results of core genes obtained by three different algorithms in Cytoscape.**
(DOC)

**S4 Table. Sankey diagram dataset of the top 20 pathways for the core targets.**
(DOC)

**S5 Table. The top 20 pathways of potential and core targets.**
(DOC)

**S6 Table. Information sheet of proteins and ligands in molecular docking.**
(DOC)

**S7 Table. Heatmap dataset of the core genes.**
(DOC)

**S8 Table. Gene expression datasets of 8 validated targets.**
(DOC)

**S9 Table. The top 20 pathways were ranked by three datasets.**
(DOC)

**S1 File. The Venn diagram dataset of triclosan and liver injury targets.**
(XLS)

**S2 File. The microarray dataset of GSE169072.**
(XLS)

## Author contributions

**Conceptualization:** Yangjia Chen.

**Data curation:** Fuzhi Liu, Yanyan Zhao.

**Formal analysis:** Fuzhi Liu.

**Methodology:** Dandan Zhu, Yuan Su.

**Project administration:** Fuzhi Liu.

**Resources:** Dandan Zhu.

**Software:** Zhaocheng Zhuang.

**Supervision:** Zhuote Tu.

**Validation:** Jingwen Wang.

**Visualization:** Zhaocheng Zhuang.

**Writing – original draft:** Fuzhi Liu, Yanyan Zhao.

**Writing – review & editing:** Fuzhi Liu, Yanyan Zhao, Yangjia Chen, Zhuote Tu.

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
