## [Decision Letter · Decision Letter 0]

14 Oct 2025

PONE-D-25-48366Mechanistic Insights into Triclosan-Induced Hepatotoxicity: A network toxicology and molecular docking approachPLOS ONE

Dear Dr. Liu,

Thank you for submitting your manuscript to PLOS ONE. After careful consideration, we feel that it has merit but does not fully meet PLOS ONE’s publication criteria as it currently stands. Therefore, we invite you to submit a revised version of the manuscript that addresses the points raised during the review process.

We look forward to receiving your revised manuscript.

Kind regards,

Yanggang Hong

Academic Editor

PLOS ONE

Journal Requirements:

Additional Editor Comments (if provided):

Network toxicology approaches, while promising, face several limitations, including methodological variability, lack of standardized validation frameworks, and challenges in integrating heterogeneous data sources. Please read and cite [PMID: 40404510].

The section on network toxicology should be expanded to provide readers with a clearer understanding of its theoretical basis and tools. For example, network toxicology constructs compound-target-pathway-disease relationships based on bioinformatics platforms, which can predict toxicity mechanisms before in vivo validation. Drawing and citing from the following literature, this section should clarify database selection, target prediction criteria, and interpretation strategies [PMID: 40317014, 39550874, 40474796, 40465018].

Reviewers' comments:

Reviewer's Responses to Questions

**Comments to the Author**

1. Is the manuscript technically sound, and do the data support the conclusions?

Reviewer #1: Yes

Reviewer #2: Yes

Reviewer #3: Yes

2. Has the statistical analysis been performed appropriately and rigorously? 

Reviewer #1: Yes

Reviewer #2: Yes

Reviewer #3: Yes

3. Have the authors made all data underlying the findings in their manuscript fully available?

Reviewer #1: Yes

Reviewer #2: Yes

Reviewer #3: Yes

4. Is the manuscript presented in an intelligible fashion and written in standard English?

Reviewer #1: Yes

Reviewer #2: Yes

Reviewer #3: Yes

5. Review Comments to the Author

Reviewer #1: No significant observation was noted in the revised manuscript. However, it is worth noting the several factors listed that are limitations to the study, especially the possibility of other endocrine disruptors and individual factors influencing the observed results. These are significant factors.

Reviewer #2: The authors should consider the chemical characteristics of triclosan* and relate its fat solubility and ability to accumulate in fat-rich tissues to its reported toxicity. They should also consider the biochemical possibility of interacting with the proteins mentioned in cellular and extracellular environments.

*Triclosan is liposoluble (fat-soluble), meaning it is almost insoluble in water but soluble in most organic solvents and alkaline solutions. Its lipophilic nature allows it to accumulate in fatty tissues and bioaccumulate in organisms, which is a major environmental and health concern. Triclosan has a lipophilic (fat-loving) character, indicated by its high log Kow value (4.8), meaning it readily dissolves in fats and oils rather than water. It is almost completely insoluble in water. It is soluble in most non-polar organic solvents and strongly basic solutions.

Reviewer #3: 1. The introduction contains several grammatical errors and awkward phrasing that hinder clarity. Sentences should be reviewed for tense consistency, article usage and overall sentence structure. A thorough language edit is recommended to improve readability and academic tone. Several ideas are repeated across different sentences for example, the health risks of TCS, its accumulation in the body, and its potential hepatotoxicity. Consider consolidating overlapping points to improve flow and reduce redundancy.

2. The introduction relies heavily on listing previously reported toxic effects of TCS without critically discussing gaps in current knowledge. It would strengthen the narrative to briefly highlight what remains unclear (e.g., molecular mechanisms of hepatocarcinogenesis) earlier in the paragraph to frame the rationale for the current study. Some statements are too broad or speculative without proper qualifiers. For instance, the claim that "TCS is an environmental endocrine disruptor and has potential health hazards to human body" should be more specific, ideally supported by mechanistic or epidemiological evidence.

3. The numbers of predicted targets (3,533 TCS-related; 2,501 liver-related; 683 overlapping) are clearly presented, but the significance of these overlaps should be interpreted. Is 683 a large overlap? Was any enrichment analysis done to evaluate the significance of this intersection?

4. GO/KEGG enrichment results are reported clearly, but some key elements are missing:

-Were adjusted p-values or FDRs used for all categories? This should be explicitly stated.

-The biological implications of the enriched pathways (e.g., how AGE-RAGE or PI3K-Akt relates to known TCS hepatotoxicity mechanisms) should be briefly discussed in the results, even if interpretation is reserved for the Discussion section. How might TP53, EGFR, AKT1, and IL6 converge within the same or related pathways? Can the discussion speculate on inter-pathway crosstalk (e.g., inflammatory and carcinogenic pathways) as a mechanistic basis for TCS hepatotoxicity?

-The term “significantly enriched” is used — what threshold was applied? (e.g., FDR < 0.05?)

5. The union of results from CytoNCA, MCODE, and CytoHubba yielded 29 targets — it would help to indicate the biological relevance of these core nodes beyond degree values (e.g., why are TP53, EGFR, and IL6 particularly important in the context of liver toxicity?).

6. Good use of comparative pathway analysis, but the ranking shifts (e.g., IL-17 moving from 17th to 2nd) are reported without much interpretation. A brief comment on why IL-17 and TNF signaling pathways might become more prominent in core targets would strengthen this section. Consider commenting on the consistency across the three analyses (potential, core, validated targets). How much overlap was observed, and is this biologically meaningful?

7. The conclusion that “TCS can spontaneously bind to each target protein” may be overstated based on docking alone, please rephrase to reflect this is a computational prediction and requires experimental validation.

8. Heatmap results are briefly mentioned. Consider reporting fold changes and statistical significance (p-values or adjusted p-values) for the genes. Only 6 genes are mentioned here (TP53, EGFR, AKT1, IL6, JUN, FN1). What happened to the others from the original set of 8? Was ESR1 or TNF not significantly changed? Please clarify.

9. Prioritize the limitations by impact — for example, the lack of experimental validation should be listed first and emphasized more strongly.Clarify the limitation around “exposure scenarios” (Line 350): Are you referring to realistic environmental concentrations, mixtures of pollutants, or bioavailability differences?You might consider adding a brief point about the limitations of public databases used (e.g., potential biases or incompleteness in SwissTargetPrediction, CTD, etc.).

6. PLOS authors have the option to publish the peer review history of their article (what does this mean? ). If published, this will include your full peer review and any attached files.

**Do you want your identity to be public for this peer review?** For information about this choice, including consent withdrawal, please see our Privacy Policy .

Reviewer #1: **Yes:** Chidi Asuzu

Reviewer #2: No

Reviewer #3: **Yes:** Amrita Basu

---

## [Author Response · Author response to Decision Letter 1]

14 Nov 2025

Dear Reviewers and Academic Editor Yanggang Hong:

Thank you for your letter and for the comments concerning our manuscript entitled "Mechanistic Insights into Triclosan-Induced Hepatotoxicity: A network toxicology and molecular docking approach" (PONE-D-25-48366). Those comments are all valuable and very helpful for revising and improving our paper, as well as the important guiding significance to our research. We have studied the comments carefully and have made corrections which we hope meet with approval. Revised portions are marked in color on the paper. The main corrections in the paper and the responds to the comments are as following:

Responds to the Editor:

1. Please ensure that your manuscript meets PLOS ONE's style requirements, including those for file naming. The PLOS ONE style templates can be found at https://journals.plos.org/plosone/s/file?id=wjVg/PLOSOne_formatting_sample_main_body.pdf and https://journals.plos.org/plosone/s/file?id=ba62/PLOSOne_formatting_sample_title_authors_affiliations.pdf.

Response: Thank you for this important reminder. We confirm that we have reviewed PLOS ONE's style requirements and have formatted our manuscript accordingly. Specific adjustments include:

File Naming: The manuscript file has been named using the journal's recommended format.

Title Page: The title page includes all required elements in the correct order and format.

Main Text Structure: The manuscript follows the standard PLOS ONE structure (Abstract, Introduction, Materials and Methods, Results, Discussion, Conclusion, References).

Figure Placement: All figures are cited in appropriate locations within the text.

Reference Formatting: References are numbered consecutively and formatted according to PLOS ONE requirements.

Supporting Information: All supporting information is clearly labeled and referenced in the manuscript.

We have used the official PLOS ONE templates to ensure complete compliance with all formatting guidelines.

Response: Thank you for highlighting PLOS ONE's policy on code sharing. We confirm that our study complies with these guidelines, as outlined below:

Regarding Author-Generated Code:

Our research utilized established, publicly available bioinformatics tools and platforms (including SwissTargetPrediction, STITCH, STRING, DAVID, Cytoscape with CytoNCA/ MCODE/ CytoHubba plugins, and CB-Dock2) for target prediction, network analysis, enrichment studies, and molecular docking. The analyses performed did not involve the creation of new, custom software or algorithms that would require code sharing. All findings are based on the standard application of these tools.

Data and Reproducibility:

To ensure full reproducibility of our results, we have provided:

All processed data files (including the final lists of potential targets, core targets, and enrichment results) as Supporting Information.

A detailed, step-by-step description of all analytical procedures and parameters in the Materials and Methods section.

All relevant accession numbers, database version numbers, and software settings required to replicate the analyses.

We affirm that our manuscript adheres to PLOS ONE's guidelines on materials and software sharing, and that the work can be fully reproduced based on the information provided in the manuscript and its supporting files.

Additional Editor Comments (if provided):

Network toxicology approaches, while promising, face several limitations, including methodological variability, lack of standardized validation frameworks, and challenges in integrating heterogeneous data sources. Please read and cite [PMID: 40404510].

The section on network toxicology should be expanded to provide readers with a clearer understanding of its theoretical basis and tools. For example, network toxicology constructs compound-target-pathway-disease relationships based on bioinformatics platforms, which can predict toxicity mechanisms before in vivo validation. Drawing and citing from the following literature, this section should clarify database selection, target prediction criteria, and interpretation strategies [PMID: 40317014, 39550874, 40474796, 40465018].

Response: We thank the Editor and Reviewer for this valuable suggestion. We have expanded the Introduction and Materials and Methods sections to provide a clearer explanation of the theoretical basis, methodology, and inherent limitations of network toxicology. Specifically, we have elaborated on our criteria for database selection, target prediction, and data interpretation strategies. As recommended, we have also cited the suggested literature to strengthen the methodological foundation of our study and to contextualize our work within the current landscape of the field. (Please see page 4, line 97 and page 4, line 114)

Responds to the Reviewers:

Reviewer #1: No significant observation was noted in the revised manuscript. However, it is worth noting the several factors listed that are limitations to the study, especially the possibility of other endocrine disruptors and individual factors influencing the observed results. These are significant factors.

Response: We agree with the reviewer that these are important limitations. We have further emphasized the potential influence of other endocrine disruptors and individual factors (such as age and gender) on hepatotoxicity outcomes in the revised Limitations section to ensure they receive appropriate attention. (Please see page 14, line 402)

Reviewer #2: The authors should consider the chemical characteristics of triclosan* and relate its fat solubility and ability to accumulate in fat-rich tissues to its reported toxicity. They should also consider the biochemical possibility of interacting with the proteins mentioned in cellular and extracellular environments.

*Triclosan is liposoluble (fat-soluble), meaning it is almost insoluble in water but soluble in most organic solvents and alkaline solutions. Its lipophilic nature allows it to accumulate in fatty tissues and bioaccumulate in organisms, which is a major environmental and health concern. Triclosan has a lipophilic (fat-loving) character, indicated by its high log Kow value (4.8), meaning it readily dissolves in fats and oils rather than water. It is almost completely insoluble in water. It is soluble in most non-polar organic solvents and strongly basic solutions.

Response: We thank the reviewer for this insightful comment. We have added a discussion on the lipophilic nature of triclosan (log Kow = 4.8), its tendency to bioaccumulate in fatty tissues, and how this property facilitates its interaction with cellular and extracellular proteins, thereby contributing to its hepatotoxicity. This addition has been incorporated into the Introduction. (Please see page 2, line 52)

Reviewer #3:

1. The introduction contains several grammatical errors and awkward phrasing that hinder clarity. Sentences should be reviewed for tense consistency, article usage and overall sentence structure. A thorough language edit is recommended to improve readability and academic tone. Several ideas are repeated across different sentences for example, the health risks of TCS, its accumulation in the body, and its potential hepatotoxicity. Consider consolidating overlapping points to improve flow and reduce redundancy.

Response: We would like to express our sincere apologies for these oversights. The Introduction has undergone comprehensive revision by a professional editor to address grammatical inaccuracies, improve syntactic structure, and ensure verb tense consistency. Repetitive content related to TCS health risks, bioaccumulation potential, and hepatotoxicity has been consolidated to enhance textual coherence and academic rigor. (Please see page 2, line 48)

2. The introduction relies heavily on listing previously reported toxic effects of TCS without critically discussing gaps in current knowledge. It would strengthen the narrative to briefly highlight what remains unclear (e.g., molecular mechanisms of hepatocarcinogenesis) earlier in the paragraph to frame the rationale for the current study. Some statements are too broad or speculative without proper qualifiers. For instance, the claim that "TCS is an environmental endocrine disruptor and has potential health hazards to human body" should be more specific, ideally supported by mechanistic or epidemiological evidence.

Response: We thank the reviewer for this constructive suggestion. We have restructured the Introduction to more clearly articulate the specific gaps in current knowledge, particularly the underexplored molecular mechanisms of TCS-induced hepatocarcinogenesis, which now serves as a stronger rationale for our study. We have also revised broad statements to be more precise and provided supporting evidence for the classification of TCS as an environmental endocrine disruptor. (Please see page 2, line 48)

3. The numbers of predicted targets (3,533 TCS-related; 2,501 liver-related; 683 overlapping) are clearly presented, but the significance of these overlaps should be interpreted. Is 683 a large overlap? Was any enrichment analysis done to evaluate the significance of this intersection?

Response: Thank you for your suggestion. The 683 predicted targets do indeed exhibit significant overlap. The primary objective of this study is to identify core targets and pathways implicated in TCS-induced liver toxicity. The large number of predicted targets facilitates subsequent screening for core targets and pathways. These findings will be detailed in the subsequent results section. Additionally, we have already presented the enrichment analysis of the 683 predicted targets in the results section.

4. GO/KEGG enrichment results are reported clearly, but some key elements are missing:

-Were adjusted p-values or FDRs used for all categories? This should be explicitly stated.

-The biological implications of the enriched pathways (e.g., how AGE-RAGE or PI3K-Akt relates to known TCS hepatotoxicity mechanisms) should be briefly discussed in the results, even if interpretation is reserved for the Discussion section. How might TP53, EGFR, AKT1, and IL6 converge within the same or related pathways? Can the discussion speculate on inter-pathway crosstalk (e.g., inflammatory and carcinogenic pathways) as a mechanistic basis for TCS hepatotoxicity?

-The term “significantly enriched” is used — what threshold was applied? (e.g., FDR < 0.05?)

Response:

We have explicitly stated in the Results section that FDR < 0.05 was used as the significance threshold for all GO and KEGG enrichment analyses. (Please see page 5, line 142)

As suggested, we have added a brief interpretation of the biological implications of key enriched pathways (e.g., AGE-RAGE, PI3K-Akt) in the Results. Furthermore, we have significantly expanded the Discussion to elaborate on how the core targets (TP53, EGFR, AKT1, IL6) converge within related pathways and to speculate on the crosstalk between inflammatory and carcinogenic pathways as a potential mechanistic basis for TCS hepatotoxicity. (Please see page 12, line 350)

5. The union of results from CytoNCA, MCODE, and CytoHubba yielded 29 targets — it would help to indicate the biological relevance of these core nodes beyond degree values (e.g., why are TP53, EGFR, and IL6 particularly important in the context of liver toxicity?).

Response: We have expanded both the Results and Discussion sections to explain the biological relevance of the core targets, with a specific focus on TP53, EGFR, and IL6, emphasizing their well-established roles in liver pathology, including apoptosis, inflammation, cell proliferation, and carcinogenesis. We also now comment on the high consistency observed across the three analytical methods (CytoNCA, MCODE, CytoHubba) as evidence of a robust and biologically meaningful selection of core targets. (Please see page 12, line 350)

6. Good use of comparative pathway analysis, but the ranking shifts (e.g., IL-17 moving from 17th to 2nd) are reported without much interpretation. A brief comment on why IL-17 and TNF signaling pathways might become more prominent in core targets would strengthen this section. Consider commenting on the consistency across the three analyses (potential, core, validated targets). How much overlap was observed, and is this biologically meaningful?

Response: We thank the reviewer for this insightful suggestion. We have added a brief interpretation in the Results section regarding the increased prominence of the IL-17 and TNF signaling pathways among the core targets, suggesting their central role in the inflammatory response underlying TCS-induced hepatotoxicity. During the refinement process from 683 potential targets to 29 core targets, the enrichment significance of these two pathways increased substantially. This enhanced prominence likely reflects their pivotal functions in key processes of TCS-induced liver injury, particularly the regulation of inflammatory responses and cellular proliferation. (Please see page 10, line 277)

We have now performed a consistency analysis across the three target sets (potential, core, and validated). The results reveal a high degree of overlap, particularly among the top-ranked targets and pathways, which strongly supports the biological relevance of our screening strategy. A detailed interpretation of this consistency has been added to the Results section. (Please see page 12, line 350)

7. The conclusion that “TCS can spontaneously bind to each target protein” may be overstated based on docking alone, please rephrase to reflect this is a computational prediction and requires experimental validation.

Response: We agree and have rephrased the conclusion in both the Results sections. The phrasing now accurately reflects that the binding is a computational prediction indicating potential binding affinity, and we explicitly state that these findings require further experimental validation. (Please see page 9, line 258)

8. Heatmap results are briefly mentioned. Consider reporting fold changes and statistical significance (p-values or adjusted p-values) for the genes. Only 6 genes are mentioned here (TP53, EGFR, AKT1, IL6, JUN, FN1). What happened to the others from the original set of 8? Was ESR1 or TNF not significantly changed? Please clarify.

Response: We have updated the Results section to include the fold-change values and statistical significance (p-values) for the six significantly altered key genes. We have also clarified that ESR1 and TNF were not significantly differentially expressed in the gene expression validation dataset (GSE169072) under the specified experimental conditions, which is now explicitly stated. (Please see page 10, line 269)

9. Prioritize the limitations by impact — for example, the lack of experimental validation should be listed first and emphasized more strongly.Clarify the limitation around “exposure scenarios” (Line 350): Are you referring to realistic environmental concentrations, mixtures of pollutants, or bioavailability differences?You might consider adding a brief point about the limitations of public databases used (e.g., potential biases or incompleteness in SwissTargetPrediction, CTD, etc.).

Response: We thank the reviewer for these crucial suggestions. We have revised the Limitations section as follows:

Reordered the limitations to list the lack of experimental validation as the primary and most significant limitation.

Clarified the "exposure scenarios" limitation to specif

---

## [Decision Letter · Decision Letter 1]

2 Feb 2026

Mechanistic Insights into Triclosan-Induced Hepatotoxicity: A network toxicology and molecular docking approach

PONE-D-25-48366R1

Dear Dr. Liu,

We’re pleased to inform you that your manuscript has been judged scientifically suitable for publication and will be formally accepted for publication once it meets all outstanding technical requirements.

Kind regards,

Ch Ratnasekhar, Ph.D.

Academic Editor

PLOS One

Additional Editor Comments (optional):

Reviewers' comments:

Reviewer's Responses to Questions

**Comments to the Author**

1. If the authors have adequately addressed your comments raised in a previous round of review and you feel that this manuscript is now acceptable for publication, you may indicate that here to bypass the “Comments to the Author” section, enter your conflict of interest statement in the “Confidential to Editor” section, and submit your "Accept" recommendation.

Reviewer #1: All comments have been addressed

Reviewer #3: All comments have been addressed

2. Is the manuscript technically sound, and do the data support the conclusions?

Reviewer #1: Yes

Reviewer #3: Yes

3. Has the statistical analysis been performed appropriately and rigorously? 

Reviewer #1: Yes

Reviewer #3: Yes

4. Have the authors made all data underlying the findings in their manuscript fully available?

Reviewer #1: Yes

Reviewer #3: Yes

5. Is the manuscript presented in an intelligible fashion and written in standard English?

Reviewer #1: Yes

Reviewer #3: Yes

6. Review Comments to the Author

Reviewer #1: The manuscript has been reviewed and it addressed every concern raised previously. I do not perceive any ethical concern with the research work. The authors response to the comments were satisfactory.

Reviewer #3: (No Response)

7. PLOS authors have the option to publish the peer review history of their article (what does this mean? ). If published, this will include your full peer review and any attached files.

**Do you want your identity to be public for this peer review?** For information about this choice, including consent withdrawal, please see our Privacy Policy .

Reviewer #1: **Yes:** Chidi Asuzu

Reviewer #3: **Yes:** Amrita Basu

---

## [Editor Report · Acceptance letter]

PONE-D-25-48366R1

PLOS One

Dear Dr. Liu,

I'm pleased to inform you that your manuscript has been deemed suitable for publication in PLOS One. Congratulations! Your manuscript is now being handed over to our production team.

Kind regards,

on behalf of

Dr. Ch Ratnasekhar

Academic Editor

PLOS One